# Theoretical Studies on the Role of Guest in α-CL-20/Guest Crystals

**DOI:** 10.3390/molecules27103266

**Published:** 2022-05-19

**Authors:** Mingming Zhou, Caichao Ye, Dong Xiang

**Affiliations:** 1College of Chemistry and Environmental Engineering, Yangtze University, Jingzhou 434023, China; 2021720391@yangtzeu.edu.cn; 2Academy for Advanced Interdisciplinary Studies, Southern University of Science and Technology, Shenzhen 518055, China; yecc@sustech.edu.cn

**Keywords:** host–guest, intermolecular interaction, electrostatic potential, CL-20, COHP

## Abstract

The contradiction between energy and safety of explosives is better balanced by the host–guest inclusion strategy. To deeply analyze the role of small guest molecules in the host–guest system, we first investigated the intermolecular contacts of host and guest molecules through Hirshfeld surfaces, 2-D fingerprint plots and electrostatic interaction energy. We then examined the strength and nature of the intermolecular interactions between CL-20 and various small molecules in detail, using state-of-the-art quantum chemistry calculations and elaborate wavefunction analyses. Finally, we studied the effect of the small molecules on the properties of CL-20, using density functional theory (DFT). The results showed that the spatial arrangement of host and guest molecules and the interaction between host and guest molecules, such as repulsion or attraction, may depend on the properties of the guest molecules, such as polarity, oxidation, hydrogen content, etc. The insertion of H_2_O_2_, H_2_O, N_2_O, and CO_2_ had significant influence on the electrostatic potential (ESP), van der Waals (vdW) potential and chemical bonding of CL-20. The intermolecular interactions, electric density and crystal orbital Hamilton population (COHP) clarified and quantified the stabilization effect of different small molecules on CL-20. The insertion of the guest molecules improved the stability of CL-20 to different extents, of which H_2_O_2_ worked best.

## 1. Introduction

As the most widely investigated high energetic compound, hexanitrohexaazaisowurtzitane (CL-20, with typical polymorphs of α-, β-, γ- and ε-) has the highest energy density [1], but it is still not widely in service due to its high sensitivity [2], phase transformations [1] and high cost. Host–guest inclusion strategy is an effective method to significantly alleviate the contradiction between high energy and low sensitivity. The development of host–guest compound explosives can solve the problems of laboriousness and risk of developing new energetic materials [3,4,5,6,7]. Bennion et al. [6] incorporated one solvate hydrogen peroxide (H_2_O_2_) molecule into the crystal system of anhydrous ε-CL-20, and obtained the CL-20 hydrogen peroxide solvate (CL-20/H_2_O_2_) for the first time. It had high crystallographic density (2.03 g·cm^−3^), high predicted detonation velocity/pressure, performed better than ε-CL-20, and had a sensitivity similar to that of ε-CL-20. Xu et al. [7] incorporated oxidizing gas molecules (N_2_O, CO_2_) into ε-CL-20 to obtain the CL-20/N_2_O and CL-20/CO_2_ complexes. CL-20/N_2_O exhibited a surprisingly high crystallographic density (2.038 g·cm^−3^ at 298 K), more thermal stability, better predicted detonation properties and lower sensitivity compared with ε-CL-20. The guest-accessible volume in α-CL-20, without being occupied by water, revealed sufficient void space to encompass some solvent molecules such as two H_2_O_2_, CO_2_ and N_2_O. Little deformation of the lattice parameters was changed after removing the water under heating/vacuum from α-CL-20 [8]. Therefore, the crystal structures of (a) CL-20, (b) CL-20/H_2_O_2_, (c) CL-20/CO_2_, (d) CL-20/N_2_O, and (e) CL-20/H_2_O remained isostructural to the hydrated α-CL-20 [6,7]. A chemical diagram and molecular structure of CL-20 are shown in Figure 1, and a planar view of the cocrystal structures are shown in Figure 2. These structures are reliable. Their original structures are the crystal structures downloaded from CCDC, and their CCDC numbers are 251409, 1495519, 771863, 1585914, and 1495521, respectively. Extensive calculation work [9,10] has demonstrated that the incorporation of small guest molecules improved the detonation performance (detonation heat, detonation pressure, detonation velocity, etc.) of the host high-energy explosive CL-20. The practical application of the host–guest explosive is always inseparable from the interaction between host and guest molecules in the system. A study of the intermolecular interaction is crucial for understanding the behavior of the complex in the actual environment and facilitating its application in practice. Meanwhile, the intermolecular interaction is the central scientific issue of energetic cocrystals [11,12,13]. Therefore, systematic studies on the comparison of interactions between the host–guest energetic complexes constructed by embedding different small molecules into the crystal lattice cavity of α-CL-20, are necessary. Further research devoted to summarizing the influence of guest molecules on the performance of α-CL-20 in order to explore more host–guest energetic complexes, is necessary.

In this paper, we will explore the potential interaction between the host and guest molecules and summarize the influence of guest molecules on the host explosive. Specifically, Hirshfeld surfaces and 2-D fingerprint plots show the intermolecular contacts. First, we examine how the electrostatic interaction energy shows the strength of the intermolecular contacts. Then, we employ ESP and vdW potential to intuitively describe the electrostatic and vdW interaction characteristics of the host and guest molecules. This analysis provides us with a general understanding of the basic character of the intermolecular interaction of this species. After that, we carefully examine the COHP analysis, charge density and difference charge density of host–guest complexes; the composition of each will be useful for detailed chemical bonding analysis. This part of the research will help us grasp the chemical bonding variation of CL-20 by embedding different small molecules. Finally, we summarize which guest type has dominant influence on the host CL-20.

## 2. Computational Methods

The Gaussian 16 (A.03) [14] program was implemented to analyze geometry optimizations and frequency using the uB97XD exchange correlation function [15] in conjunction with the def2-TZVP basis set [16]. The process of geometry optimization is the process of obtaining reasonable structure. In order to obtain a reliable WFN file, it was converted from CIF file format. Then, Multiwfn 3.7 code, developed by one of the authors of this paper in [17], was performed to analyze the ESP, vdW potential, non-covalent interaction (NCI) map, contour map of electrostatic potential, electrostatic interaction energy, IRI and isosurface map of electron density on the basis of the optimized geometry. Visual Molecular Dynamics (VMD) software [18] was rendered to analyze isosurface maps of various real space functions based on the files exported by Multiwfn. 

A freely available software, CrystalExplorer [19], was not only applied to visualize ab initio molecular ESPs mapped on Hirshfeld surfaces [20] or isosurfaces of the electron density, but was also used to calculate quantum-mechanical properties of molecules [21]. In the region of the hydrogen bonds, the isosurfaces overlap significantly, whereas Hirshfeld surfaces touch, and quite clearly demonstrate the way in which complementary electropositive (blue) and electronegative (red) regions of adjacent molecules come into contact in such an interaction [22]. The intermolecular interactions in crystals can be directly observed by electrostatic potentials mapped on Hirshfeld surfaces. The intermolecular interactions can be analyzed quantitatively and qualitatively by two-dimensional mapping [23,24] in a convenient color plot. The intermolecular contacts were explored by the points on the surface which were defined by the distances to the nearest atoms outside, d_e_, and inside, d_i_ [25]. During the theoretical investigations by CrystalExplorer, the chosen part of crystal structures was automatically selected by CrystalExplorer software from the CIF files. The chosen part of crystal structures was periodic for the crystal structures.

Bader’s QTAIM [26] method and DFT calculations with Critic2 [25] were used to investigate the intra- and inter-molecular interaction strength. The core and valence electron densities and difference charge density of each crystal were obtained from DFT calculations implemented by VASP [27]. During the theoretical investigations by VASP, the chosen part of crystal structures was selected by the nearest two neighbors. Furthermore, although LOBSTER was originally designed with interfaces to handle only wavefunctions from VASP [27], while using LOBSTER, the crystal orbital Hamilton population (COHP) analysis was demonstrated and reported in [28]. LOBSTER is a multiplatform tool that is written in object-oriented C^++^ and parallelized using OpenMP. It employs Boost libraries [29] in addition to the highly efficient Eigen library [30].

## 3. Results and Discussion

### 3.1. The Intermolecular Contacts of Host and Guest Molecules

Usually, two interacting molecules stack in a special direction due to electrostatic attraction. The magnitude of the electrostatic attraction depends on the closer contact of the positive and negative ESPs [30,31]. From the surface minima and maxima sites (shown in Figure 3), the outermost distribution are minima of CL-20. The minima sites are evenly distributed in the extension space corresponding to O atoms of N-NO_2_ fragments. This demonstrates that CL-20 more easily forms H-bonds with other molecules at the minima sites. Therefore, the distribution of guest molecules is affected by the ESP-mapped molecular vdW surface. When there are no hydrogen atoms in the guest molecules, such as N_2_O and CO_2_, the distribution of host and guest molecules is very different to that of H_2_O and H_2_O_2_. For CO_2_, there is perfect linear symmetry; CO_2_ is arranged diagonally along the two N-O bonds, with the oxygen atom closer to the CL-20 fragment. However, the minima sites are around the O atom of CO_2_. Therefore, CO_2_ and CL-20 fragments show mutual repulsion. For N_2_O, the O atom is closer to the CL-20 fragment, too. The mutual repulsion causes the CL-20 to undergo large structural changes with the large deflection of one of the NO_2_ fragments. One minima and maxima around the O atom around N_2_O causes the distribution of CL-20/N_2_O and CL-20/CO_2_ to be different. The different oxidizability of H_2_O and H_2_O_2_ causes the different distribution of CL-20/H_2_O and CL-20/H_2_O_2_. The distance between the CL-20 fragment and H_2_O_2_ is closer than that of other complexes of host and guests, as shown as Figure 3.

To obtain a better understanding of the host–guest driven inclusion behavior between CL-20 and guest molecules, the intermolecular interactions [4,32] of single crystals were studied by freeware of Hirshfeld surfaces, as shown in Figure 4. In the Hirshfeld surface analysis, the red and blue areas represent the probability of close and far contact with external molecules, respectively. The red regions arranged in the oxygen and hydrogen atoms are shown in Hirshfeld surfaces. This implies the main intermolecular interactions contacts of CL-20 are focused on O and H atoms. The reaction sites of the H atom are consistent with the conclusion of Figure 3. The number of red regions (i.e., close contacts) is the same for the CL-20 fragment of the five substances. However, the sites of the red regions for the host–guest complex are distinctly different than for CL-20. The red regions around the guest molecules are much more obvious than around the host. The intensity of the red areas for guests decreases in this order: H_2_O_2_, H_2_O, N_2_O, and lastly, CO_2_. This demonstrates that CL-20 is much more stable after incorporating guest molecules. This conclusion is consistent with the finding that CL-20/N_2_O has higher thermal stability and lower impact sensitivity than CL-20 [32]. The d_norm_ Hirshfeld surfaces for CL-20, CL-20/N_2_O, CL-20/H_2_O, and CL-20/H_2_O_2_ resemble a whole shape. However, it is quite different for CL-20/CO_2_. The Hirshfeld surface is significantly different, in that it divides into two parts. The sites of the red regions for the CL-20 fragment are obviously biased to where the CO_2_ molecule is located. The different results may be due to the different polarity of the guest molecules.

The 2-D fingerprint plot directly demonstrates the intermolecular interactions of internal and external distances of atoms from the surface. It is possible to show the range of structures by the changes in the fingerprint plots while adding the guest into the host explosive. We applied this tool to show the intermolecular interaction variation after insertion of the guest molecules in CL-20. The graph in Figure 5 changes as different guests are embedded; there is a noticeable decrease in symmetry about the x/y diagonal in the following order: CO_2_, H_2_O, N_2_O, and lastly, H_2_O_2_, especially for the fingerprint of CL-20/H_2_O_2_. For pure CL-20, Figure 6 shows that intermolecular interactions are governed by O…O contacts. The O…O interactions that usually appear in energetic crystal [33,34,35], can be readily understood by the internal moieties of CL-20 being dominated by the O atom. This is one of the factors that determine the high energy release of CL-20. The O atoms with high negative ESPs (shown in Figure 1) to form O…O contacts can lead to a big electrostatic repulsion. This is not conducive to the stability of CL-20. The steric in the cage and rings is also shown in the plots of IRI and RDG isosurface. The other two main intermolecular interactions are H…O and O…H contacts. The existence of hydrogen bonds is beneficial to the stability of CL-20. The O…O contacts percentage contribution to the Hirshfeld surface decreases in the order of: CL-20 (44.2%), CL-20/CO_2_ (42.4%), CL-20/H_2_O_2_ (37.6%), CL-20/N_2_O (36.8%), and CL-20/H_2_O (35.9%). This implies that the repulsion in the cage and rings may decrease with the same order. The intuitive exhibition of repulsion in the cage and rings are shown in the plots of the NCI isosurface. The O…H and H…O contacts percentage contribution of CL-20/H_2_O_2_ is larger than for CL-20. By combining the decreasing O…O contacts and the increasing HBs, it can be inferred that CL-20/H_2_O_2_ is more stable than CL-20.

The inter-fragment interaction between all the defined fragments (one fragment is the host CL-20, the other fragment is one of the small guest molecules) can easily be recognized by the electrostatic interaction energy. The electrostatic interaction energies between guest molecules and the selected atoms of CL-20 are shown in Table 1, with a calculation of which are closer to the guest molecules. The most important contribution to the attraction is the electrostatic interaction of the O…H, O…N and O…C contacts; this result is easy to understand since O is the acceptor atom of these contacts. However, even though there were rejections between O…O contacts and O…N contacts, the fragments between N_2_O, H_2_O_2_, H_2_O and CL-20 attract each other. This contributes greatly to the sum of the electrostatic effect with binding (−0.14, −8.93, −0.11 kJ·mol^−1^). Meanwhile, the attraction between H_2_O_2_ and CL-20 is very obvious and cannot be ignored, as the binding energy is much less than 0. This may be because of the different electrostatic potential (ESP) and van der Waals (vdW) potential of CL-20/H_2_O_2_ contrary to the other complexes. However, the mutual repulsion of the fragment CO_2_ and CL-20 contributes greatly to the sum of the electrostatic effect without binding (0.03 kJ·mol^−1^). Therefore, the CL-20 fragment combined with N_2_O, H_2_O_2_, H_2_O fragments may be a whole molecule. The CL-20 fragment with CO_2_ may be two fragments. This conclusion confirms the two parts of the Hirshfeld surface for CL-20/CO_2_, as shown in Figure 4. The two different kinds of interaction between CL-20/N_2_O, CL-20/H_2_O_2_, CL-20/H_2_O and CL-20/CO_2_ may be caused by the polarity of the guest molecules.

### 3.2. Electrostatic and vdW Interaction Characteristics of Host and Guest Molecules

The electrostatic potential (ESP) on the molecular vdW surface was used to study and predict intermolecular interaction, for example, the information of the close contact site, structure property and special hydrogen bonding [36,37,38], which is usually employed to study the molecular packing in cocrystals [39]. Therefore, it is very useful to investigate the important interaction between the host explosive and small guest molecules. The ESP-mapped vdW surface, in addition to the surface extrema of CL-20 and its host–guest complexes, are shown in Figure 7, and their surface areas are plotted as shown in Figure 8.

It can be seen that oxygen atoms of NO_2_ have negative surface potential, while C-H and cage have positive surface potential as demonstrated in Figure 7. The O of CO_2_ is closer to CL-20 and possesses negative ESP (−13.7 kcal·mol^−1^). The ESP of the CL-20 fragment, which is close to CO_2_, is also negative (−19.54, −18.87 kcal·mol^−1^). This determines that the two molecules are mutually exclusive. Of the two negative ESP for O atoms of N_2_O, one (−16.18 kcal·mol^−1^) is inter-attraction with positive ESP (44.63, 43.74 kcal·mol^−1^) of CL-20. The other (−19.38 kcal·mol^−1^) is mutually repulsive with negative ESP (−18.08, −19.42 kcal·mol^−1^) of CL-20. The equilibrium of the two forces determines the distribution of the CL-20 fragments and N_2_O. The two hydrogen atoms possess positive ESP (46.83 and 47.31 kcal·mol^−1^), corresponding with the negative ESP (−14.46 and −19.68 kcal·mol^−1^) of the CL-20 fragment. The H…CL-20 contacts are of mutual affinity. Between the two negative ESP, there are many higher positive ESP relatively far from H_2_O. The H_2_O…CL-20 contacts are mutually repulsive. The balance of the two forces determines the distribution of the CL-20 fragments and H_2_O. In the configuration of CL-20/H_2_O_2_, the negative surface potential of CL-20 (−25.40 kcal·mol^−1^) overlaps with the positive surface potential of H in H_2_O_2_ (43.8 and 46.02 kcal·mol^−1^), indicating that noncovalent CL-20…H_2_O_2_ bonds are formed at the same time. The interaction between CL-20 and H_2_O_2_ is bigger than other host and guest molecules. The values of the electrostatic interaction energy between CL-20 and the guest molecules are shown in Table 1.

Figure 8 illustrates the characteristic of ESP distribution of CL-20 and the additional guest parts in the CL-20 unit cell, respectively. For ESP distribution of CL-20, the positive part mainly arises from the positively charged C-H carbon atoms. The remarkable positive and negative ESP value are small areas, corresponding to the regions closed to the global ESP minimum (−18.5 kcal·mol^−1^) and maximum (83.5 kcal·mol^−1^), respectively. The whole CL-20 surface partition is affected by the different guest molecules. For the CL-20, the vdW surface area is 310.16 Å^2^. By comparing the experimental data, the proportion of ESP distribution of the CL-20 fragment occupies, in order: 81.4%, 84.1%, 86.6%, and 90.7% of the overall surface for CL-20/N_2_O, CL-20/CO_2_, CL-20/H_2_O, and CL-20/H_2_O_2_, respectively. It shows that the H_2_O_2_ guest has the least effect on the surface area of host–guest cell. CL-20/N_2_O has the biggest surface area. This may be caused by the polarity of whole host–guest complex. The greater the polarity, the greater the ESP surface area. It can be seen from the graph that there is a large portion of the CL-20 molecular surface having a small ESP value, namely, from −15.5 to 8.5 kcal·mol^−1^. The main distribution area widens from −18.57 to 11.15, −19.32 to 10.96, −15.5 to 14.8, and −17.54 to 27.04 kcal·mol^−1^, for CL-20/H_2_O_2_, CL-20/H_2_O, CL-20/CO_2_, and CL-20/N_2_O, respectively. The ESP surface area proportion of the CL-20 fragment can determine the degree of broadening order of the main distribution area. When the H_2_O fragment embeds into the cell of CL-20, the ESP distribution on the vdW surface fluctuates most dramatically. The higher negative ESP value of CL-20/H_2_O is the largest, while the extreme values vary little for the other complexes, compared with CL-20. This indicates that CL-20/H_2_O more easily forms H-bonds with other molecules than CL-20.

From the graph for CL-20 (Figure 9a), it is clear that the C and H atoms form bonds. The C-H fragments are overall positively charged because they largely intersect solid contour lines of the vdW surface close to the two C-H fragments. This shows that the C-H fragments are surrounded by positive value lines, and suggests that the C-H segments are more susceptible to electrophilic reactions that are more stable when they receive hydrion. While embedding different small molecules into the crystal lattice cavity of α-CL-20, the symmetry of the contour map of electrostatic potential is broken, and the contour lines become chaotic. This may be because the symmetry of CL-20 is broken. This shows that all host-guest complexes contribute to electrophilic reactions in the same manner as that of CL-20. However, the reaction sites may be a little changeable for the different intersect regions. The combining capacity with hydrion, decided by the maximum of energy data, is in the following order: CL-20/H_2_O (94.54 eV), CL-20 (78.14 eV), CL-20/CO_2_ (55.68 eV), CL-20/N_2_O (48.55 eV), and CL-20/H_2_O_2_ (11.50 eV). This implies that CL-20/H_2_O_2_ may be the most stable complex, while the CL-20/H_2_O may be the liveliest complex.

The color-filled NCI isosurface not only demonstrates where weak interaction occurs, but is also an intuitive presentation of their interaction—such as repulsion or attraction—and their magnitude. We can identify different types of regions by simply examining their colors. Recalling the color scale bar shown previously in Figure 10A, more blue implies a stronger attractive interaction. The elliptical slab between the oxygen and hydrogen atoms shows green color in Figure 10, so we can conclude that there exists a hydrogen bond, but not a very strong one. The yellow circle demonstrates the vdW interaction region, which shows that the electron density in this region is low. Obviously, the regions at the center of the cage and rings correspond to strong steric interaction, since they are filled by red. This result explains the relatively low stability of CL-20. The configuration of the CL-20 fragment changes significantly in the host–guest complex, due to the appearance of the vdW interaction region between the two NO_2_ fragments. The configuration caused by the repulsion of guest molecules and NO_2_ prepares enough space to accommodate guest molecules. The red color becomes lighter and the shape becomes thinner at the center of the cage in the host–guest complex. This demonstrates that the cage of CL-20 fragments in the host–guest complex is more stable than the CL-20. This conclusion is consistent with the finding that the CL-20/N_2_O has higher thermal stability and lower impact sensitivity than CL-20 [39,40].

Figure 10B focuses on demonstrating the interaction of the host and guest molecules. The isosurface values for CL-20/N_2_O, CL-20/H_2_O, and CL-20/H_2_O_2_ are 0.9 a.u. except for CL-20/CO_2_, which is 1.1 a.u. The interaction regions of H_2_O…CL-20, N_2_O…CL-20, and CO_2_…CL-20 contacts are green color, so the hydrogen bonds are not very strong. The strength of the interaction decreases in the order of: CL-20/H_2_O, CL-20/N_2_O, and CL-20/CO_2_, in the area of the interaction region. The interaction between CO_2_ and CL-20 is relatively small. These results are consistent with the electrostatic interaction energy between CO_2_ and CL-20. The interaction regions of H_2_O_2_…CL-20 contacts are bluish on the ends and red in the middle. The hydrogen bond and the repulsion between oxygen atoms are very strong. This strong hydrogen bond of H_2_O_2_…CL-20 contacts may be the reason for the least effect on the surface area. The position of H_2_O_2_ in the cell decides the location of the equilibrium position of the two forces.

IRI is not only able to reveal the weak interaction region, but also shows the steric effect within the cage and rings of CL-20 [8]. There are four obvious spikes below the horizontal isosurface line of RDG = 0.9, and the spikes can be classified into three types for CL-20 and its complexes [41]. The spike at about −0.275 a.u. indicates the existence of hydrogen bonds. At about −0.02 a.u. and 0.02 a.u., there are two spikes, which demonstrate that the complexes have vdW interaction. The steric effect in the cage and ring is displayed when the spike appears at about 0.275 a.u. This phenomenon explains the instability of CL-20. The influence of small guest molecules on the intramolecular interaction of CL-20 is shown in Figure 11a. In addition, the strength of the weak interaction has a positive correlation with electron density in the corresponding region. The IRI plots of CL-20 and CL-20/H_2_O_2_ are similar. This phenomenon indicates that the addition of H_2_O_2_ has little effect on the intramolecular reaction of CL-20. However, the electron density of CL-20/H_2_O, CL-20/CO_2_, and CL-20/N_2_O are rarer than of CL-20 when λ_2_ < 0. This displays that the addition of H_2_O, CO_2_, and N_2_O weakens the intramolecular forces and intermolecular hydrogen bonds. Table 1 demonstrates that the summation of H…Guest electrostatic interaction energy of CL-20/H_2_O, CL-20/CO_2_ and CL-20/N_2_O is much less than one H…H_2_O_2_ (−3.77 kJ·mol^−1^) electrostatic interaction energy. It also shows that the addition of H_2_O, CO_2_, and N_2_O weakens the intermolecular hydrogen bonds.

A comparison of molecular polarity index values reveals that the guest molecules with greater polarity such as CL-20/H_2_O (19.56 kcal·mol^−1^), CL-20/N_2_O (19.53 kcal·mol^−1^) and CL-20/H_2_O_2_ (19.5 kcal·mol^−1^) correspond to the macropolar host–guest complex more than CL-20 (18.61 kcal·mol^−1^). CL-20/CO_2_ (14.95 kcal·mol^−1^) has the smallest polarity. The polarities of CL-20 and its host–guest complexes are rather high, since their MPIs are higher than benzene (8.4 kcal·mol^−1^) [42], which possesses the common unsaturated hydrocarbons as determined by experimental chemists. This further implies that the strength of the electrostatic interaction between the CL-20 and its host–guest complex should be fairly strong. The stronger the electrostatic interaction is, the more stable the explosive is. The results are consistent with the conclusion shown by the color-filled NCI isosurface.

Owing to the relatively low polarity of CL-20/CO_2_, the vdW interaction [43] of CL-20/CO_2_ is likely to be very important with regard to its electrostatic interaction in intermolecular complexation. In Figure 12, the green isosurface represents the region where vdW is negative. The guest molecules tend to be attracted to the green region due to the driven force of dispersion attraction. The region close to the nuclei is fully enclosed by blue isosurface, indicating that the exchange repulsion potential dominates the vdW potential in this area. The surfaces of the vdW potential show that CL-20/H_2_O_2_ is a whole, while other host–guest complexes are comprised of two parts. This contributes to the closer distance between CL-20 and H_2_O_2_. This is also intuitively shown in the color-filled NCI isosurface. It further explains that the CL-20/H_2_O_2_ is the special existence for the other complex.

### 3.3. Effect on the Chemical Bonding of CL-20 by the Small Molecules

The electronic structure, shown in Figure 13, and the charge, change [44] before and after embedding different small molecules into the crystal lattice cavity of α-CL-20. The yellow isosurface (0.001 a.u.) represents the region in which the electron density of the bonds increases. As shown in Figure 13a, it is obvious that electron density shifts from C-H fragments toward N-NO_2_ fragments to strengthen the bonding energy. The electrons mainly accumulate in the branch chain of CL-20. The electronegativity of O atoms of the nitro branch chain is larger than that of N and C atoms, causing the electron cloud density of the entire cage structure to be biased towards O atoms, thus presenting greater electronegativity. The electronegativity of N atoms is smaller than that of O atoms, and the electronegativity of N atoms is larger than that of the cage structure. This demonstrates that the N-NO_2_ bonds are more active than the N-O bonds. This is consistent with the conclusion that CL-20 has only one distinct initial decomposition channel homolysis of the N-NO_2_ bond. The cages of other atoms are electropositive. The positive electricity compared with the smaller distance between atoms by the cage structure, further intensifies the mutually repulsive force of the cage structure. This is an important factor in determining the instability of CL-20. The result corresponds with the incidental phase transformations [45,46], and the cage collapse with the C-N bonds rupture [47]. Table 2 lists the charge density of each atom for CL-20 and its complexes. The variation tendency of charge density between CL-20/CO_2_ and CL-20/N_2_O are similar, while it is slightly different to CL-20. The variation tendency of charge density between CL-20/H_2_O and CL-20/H_2_O_2_ is similar. However, the variation trend is very different from CL-20. This indicates that CO_2_ and N_2_O have little effect on the charge density of CL-20, while H_2_O and H_2_O_2_ have a larger effect on the charge density of CL-20. This difference may depend on whether the guest molecule contains hydrogen atoms.

The differential charge densities of CL-20 and its complexes are shown in Figure 14. For the H_2_O_2_ embedded in CL-20, the charge density variation of CL-20 is more obvious than for the other complexes. The charge density variation of the three NO_2_ fragments close to H_2_O_2_ is remarkable. The one NO_2_ fragment away from the inserted H_2_O_2_ obtains more electrons than the two NO_2_ fragments close to the H_2_O_2_. For the two NO_2_ fragments, the two O atoms adjacent to the H atoms of H_2_O_2_ lose electrons. Therefore, the electrostatic interaction between the host and guest molecules is enhanced, which results in a decrease in the distance, and an increased intermolecular interaction. This is consistent with the charge density presented in Table 2. For the H_2_O implant in CL-20, the charge density variation of the two NO_2_ fragments close to H_2_O is remarkable. The O atom adjacent to the H atom of H_2_O loses electrons. However, the O atom of H_2_O rejects the O of NO_2_. Therefore, the distance between the H_2_O and CL-20 is decided by the balance of two forces. For the N_2_O implant in CL-20, the charge change of the two NO_2_ fragments close to H_2_O is small. The O atom adjacent to the N atom of N_2_O loses electrons. Meanwhile, the O atom of N_2_O adjacent to the O of NO_2_ loses electrons, however, the interaction is very small. This demonstrates that the distance between N_2_O and CL-20 is a little further than that of CL-20/H_2_O_2_ and CL-20/H_2_O. For the CO_2_ incorporated in CL-20, the electrons of CL-20 show little change. Only the O atoms of CO_2_ obtain electrons from the C atoms of CO_2_. The electrostatic interaction between CO_2_ and CL-20 is unchanged, which results in a large distance, as also shown in Figure 4 and Figure 10. As a result, the charge density of CL-20 changes obviously with the H_2_O and H_2_O_2_ insertions, although the charge density of CL-20 changes little for the N_2_O and CO_2_ insertions. The results also indicate that the distribution of the electrons transferred to the CL-20 is destroyed in different degrees due to the different guests. This demonstrates that the charge density variation is decided by the hydrogen contained by, and the oxidability of, guest molecules.

The COHP technique can be qualitative and correctly describes negative (i.e., bonding) and positive (i.e., antibonding) contributions [48,49] of chemical bonding with band-structure energy, as shown in Figure 15. The larger the value of the area above 0 minus the area below 0, the more bonds there are. The value of -COHP decreases according to the N-O of NO_2_, N-NO_2_, C-N and C-C, as shown in Table 3. This demonstrates the cage is relatively more unstable than other chains of CL-20. Where the distance between two atoms exceeds 1.6 Å, the value of their -COHP is positive, which shows that the intermolecular interactions are not negligible for CL-20. This demonstrates the relatively unstable of CL-20.

In order to visually and simply show the effect of small guest molecules on the COHP of CL-20, we divided the bonds into two types. The first type is cage bonds such as C-C bonds and C-N bonds. The second type is branched chain bonds such as C-H bonds, N-N bonds and N-O bonds. For CL-20/2H_2_O_2_, the -COHP of C-C bonds is significantly higher. It is a little higher for CL-20/H_2_O. However, it is lower relative to CL-20/CO_2_ and CL-20/N_2_O. The -COHP of C-N bonds is substantially higher for CL-20/2H_2_O_2_. It is a little higher for CL-20/H_2_O. However, it is lower relative to CL-20/CO_2_ and CL-20/N_2_O. The variation tendency of COHP demonstrates that the cage structure of CL-20 is made more stable by embedding H_2_O_2_ and H_2_O. However, the cage structure of CL-20 is made more unstable by embedding CO_2_ and N_2_O. The -COHP of C-H bonds is larger than of CL-20. The increase for CL-20/2H_2_O_2_ is significant. It shows that C-H bonds are more stable after adding guest molecules. For H_2_O_2_, CO_2_, N_2_O and H_2_O embedded into the crystal lattice cavity of α-CL-20, the -COHPs of N-O bonds and N-N bonds are smaller than for CL-20. This shows that branched chains may be made much livelier by embedding small molecules. These results are inconsistent with those in the literature. However, it does suggest that guest molecules containing hydrogen contribute more to the stability of CL-20.

## 4. Conclusions

We conducted a systematic and in-depth theoretical exploration and comparison of the intermolecular contacts, intermolecular interaction characteristics and chemical bonding analysis of the high-energy explosive CL-20 and the new host–guest explosives CL-20/H_2_O_2_, CL-20/H_2_O, CL-20/CO_2_ and CL-20/N_2_O. The main findings and conclusions are summarized as follows:(1)The d_norm_ Hirshfeld surfaces, 2-D fingerprint plot and individual atomic contact percentage contribution demonstrate that the cage of CL-20 fragments in host–guest complexes are more stable than CL-20. The electrostatic interaction energy shows that CL-20/H_2_O_2_ possesses stronger hydrogen bonds and stronger mutual attraction between host and guest molecules than other complexes. The specific distribution of host and guest molecules are affected by the polarity and oxidizability of the guest molecules;(2)The electrostatic interaction characteristics of host and guest molecules shows that overlapping surface potential only occurs for CL-20/H_2_O_2_. The interaction between CL-20 and H_2_O_2_ is biggest, while CL-20/H_2_O_2_ is the most stable complex compared with other complexes. The surfaces of van der Waals (vdW) potential shows that CL-20/H_2_O_2_ appears as a whole, while other host–guest complex consists of two parts. The potentials demonstrate that CL-20/H_2_O_2_ is the special existence for the other complex;(3)The charge density of CL-20 changes obviously with H_2_O and H_2_O_2_ insertion, although the charge of CL-20 changes little with N_2_O and CO_2_ insertion. The variation tendency of COHP demonstrates that the cage structure of CL-20 is made more stable by embedding H_2_O_2_ and H_2_O. However, the phenomenon is reversed by embedding CO_2_ and N_2_O. The charge density variation and stability of the host–guest complex is affected by the hydrogen content of the guest molecules.

The results of this study revealed that the guest H_2_O_2_ small molecule played a certain stable role for CL-20. For the synthesis of new energetic materials with host–guest inclusion strategy, we investigated new guest molecules by referencing properties such as geometry configuration, oxidability, polarity and hydrogen content of H_2_O_2_. Our results provide fundamental insight into the roles of guest molecules in host–guest crystals and may be helpful for the formation of new host–guest energetic materials by incorporating appropriate species of small molecules into crystal lattice voids.

## Figures and Tables

**Figure 1 molecules-27-03266-f001:**
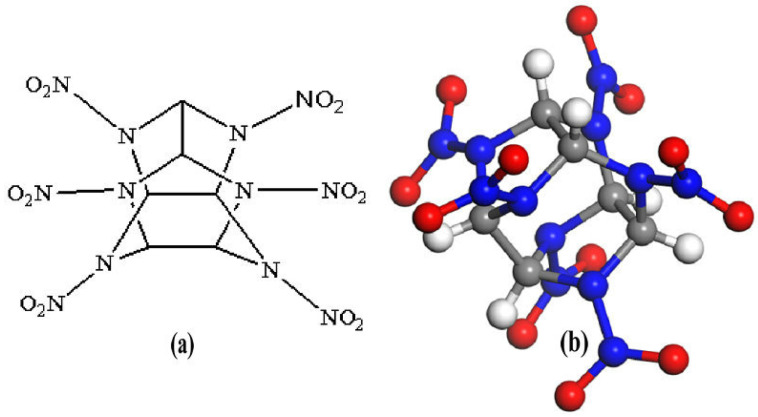
(**a**) chemical diagram of CL-20, (**b**) molecular structure of a CL-20 molecule corresponding to the chemical diagram.

**Figure 2 molecules-27-03266-f002:**
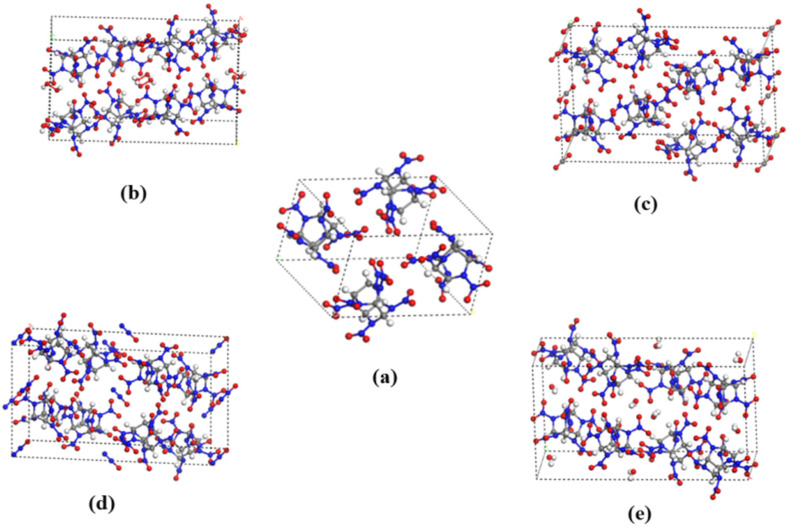
The unit cell: (**a**) CL-20, (**b**) CL-20/H_2_O_2_, (**c**) CL-20/CO_2_, (**d**) CL-20/N_2_O, and (**e**) CL-20/H_2_O. Gray, blue, red, and white spheres stand for carbon, nitrogen, oxygen and hydrogen atoms, respectively.

**Figure 3 molecules-27-03266-f003:**
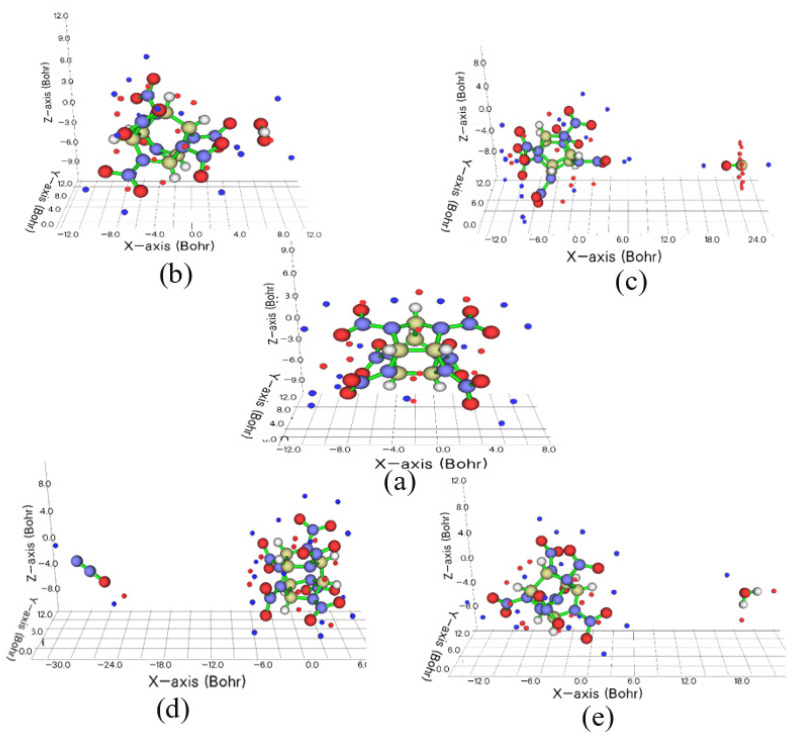
The surface minima and maxima: (**a**) CL-20, (**b**) CL-20/H_2_O_2_, (**c**) CL-20/CO_2_, (**d**) CL-20/N_2_O, and (**e**) CL-20/H_2_O. Red and blue spheres represent the position of maxima and minima, respectively.

**Figure 4 molecules-27-03266-f004:**
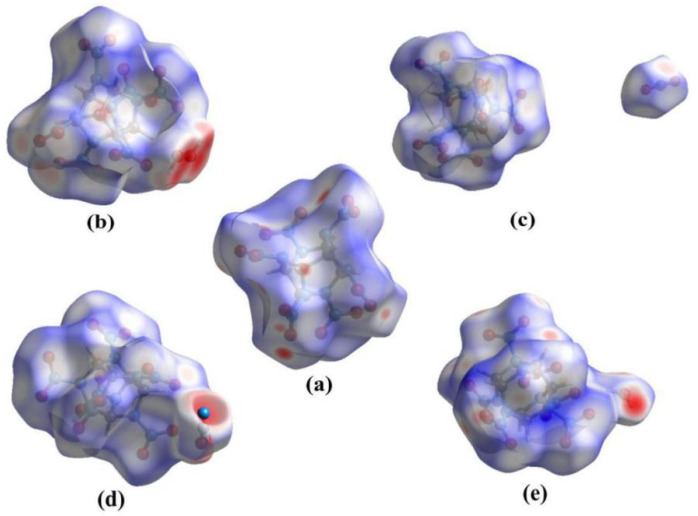
Hirshfeld surfaces of the CL-20 molecule and its four host–guest complexes showing intermolecular contacts: (**a**) CL-20, (**b**) CL-20/H_2_O_2_, (**c**) CL-20/CO_2_, (**d**) CL-20/N_2_O, and (**e**) CL-20/H_2_O.

**Figure 5 molecules-27-03266-f005:**
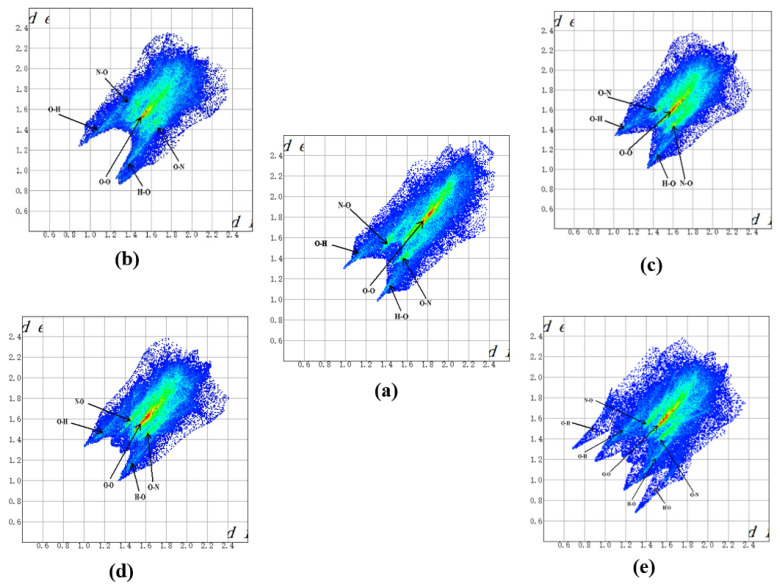
Two-dimensional fingerprint plots of close interatomic contacts of CL-20 molecules in pure form and in four host–guest complexes: (**a**) CL-20, (**b**) CL-20/H_2_O_2_, (**c**) CL-20/CO_2_, (**d**) CL-20/N_2_O, and (**e**) CL-20/H_2_O.

**Figure 6 molecules-27-03266-f006:**
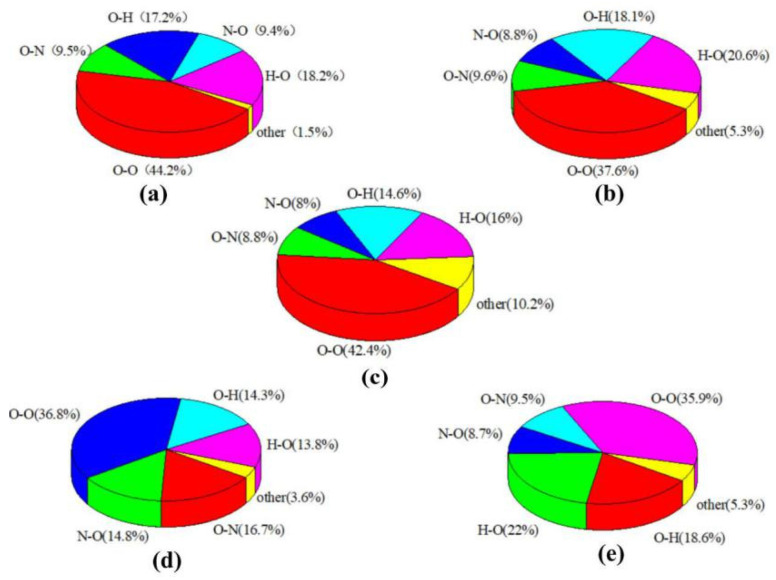
The individual atomic contact percentage contribution to the Hirshfeld surface of CL-20 and its complexes: (**a**) CL-20, (**b**) CL-20/H_2_O_2_, (**c**) CL-20/CO_2_, (**d**) CL-20/N_2_O, and (**e**) CL-20/H_2_O.

**Figure 7 molecules-27-03266-f007:**
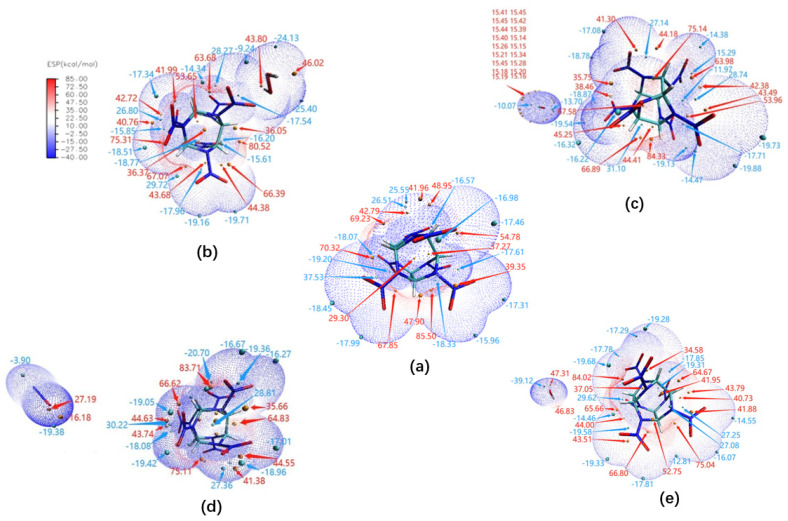
ESP—mapped molecular vdW surface of: (**a**) CL-20, (**b**) CL-20/H_2_O_2_, (**c**) CL-20/CO_2_, (**d**) CL-20/N_2_O, and (**e**) CL-20/H_2_O. The unit is in kcal·mol^−1^. Surface local minima and maxima of ESP are represented as green and orange spheres, respectively. The global minimum and maximum are labeled by blue and red colors, respectively.

**Figure 8 molecules-27-03266-f008:**
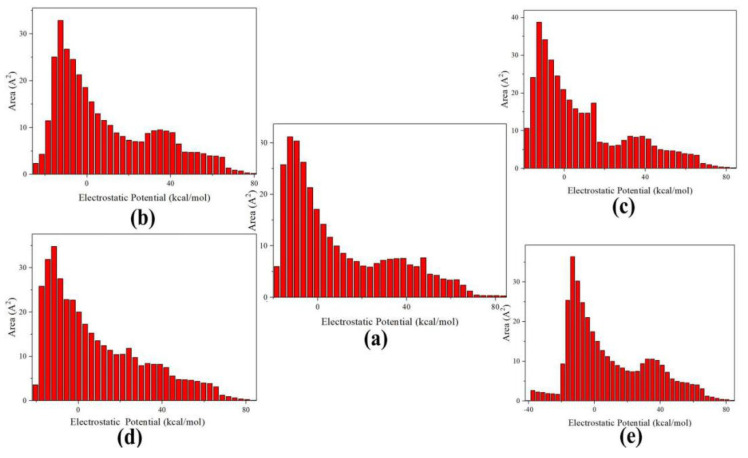
Surface area in each ESP range on the vdW surface of: (**a**) CL-20, (**b**) CL-20/H_2_O_2_, (**c**) CL-20/CO_2_, (**d**) CL-20/N_2_O, and (**e**) CL-20/H_2_O.

**Figure 9 molecules-27-03266-f009:**
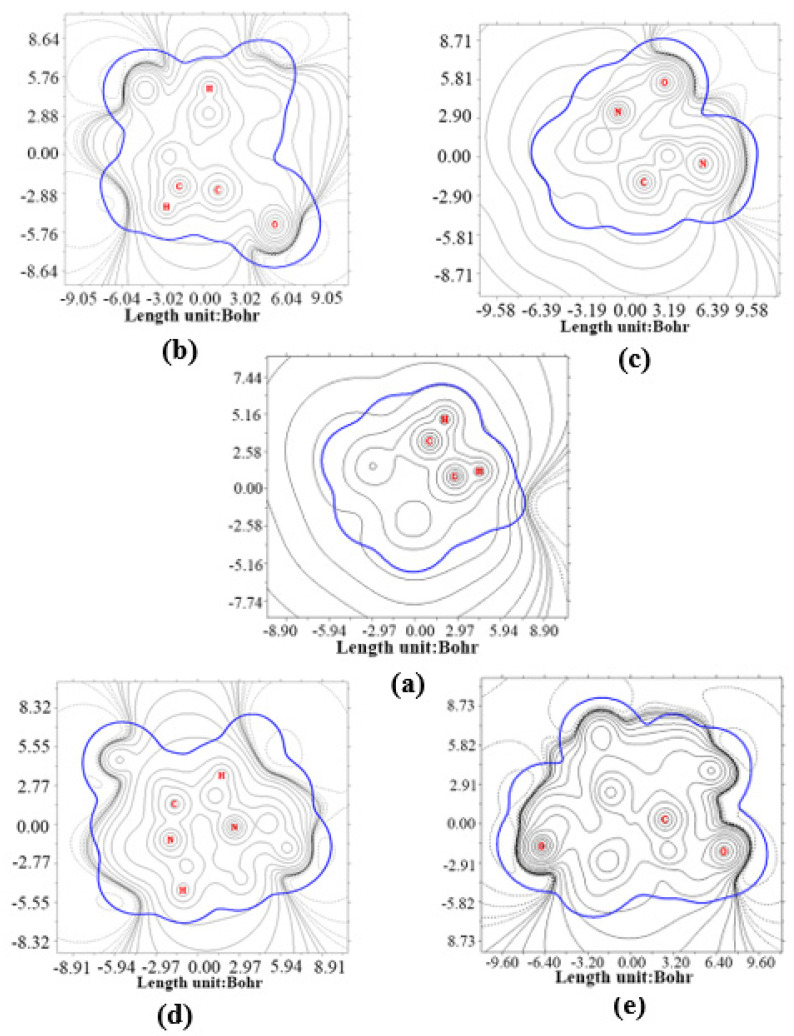
Contour map of electrostatic potential of CL-20 and it host–guest complex:(**a**) CL-20, (**b**) CL-20/H_2_O_2_, (**c**) CL-20/CO_2_, (**d**) CL-20/N_2_O, and (**e**) CL-20/H_2_O. The bold blue line corresponds to the vdW surface (isosurface of electron density = 0.001 a.u., as defined by R. F. W. Bader). The solid and dashed lines represent the region having positive and negative values of ESP, respectively. The isovalues start at ± 0.001 a.u., and double at each step.

**Figure 10 molecules-27-03266-f010:**
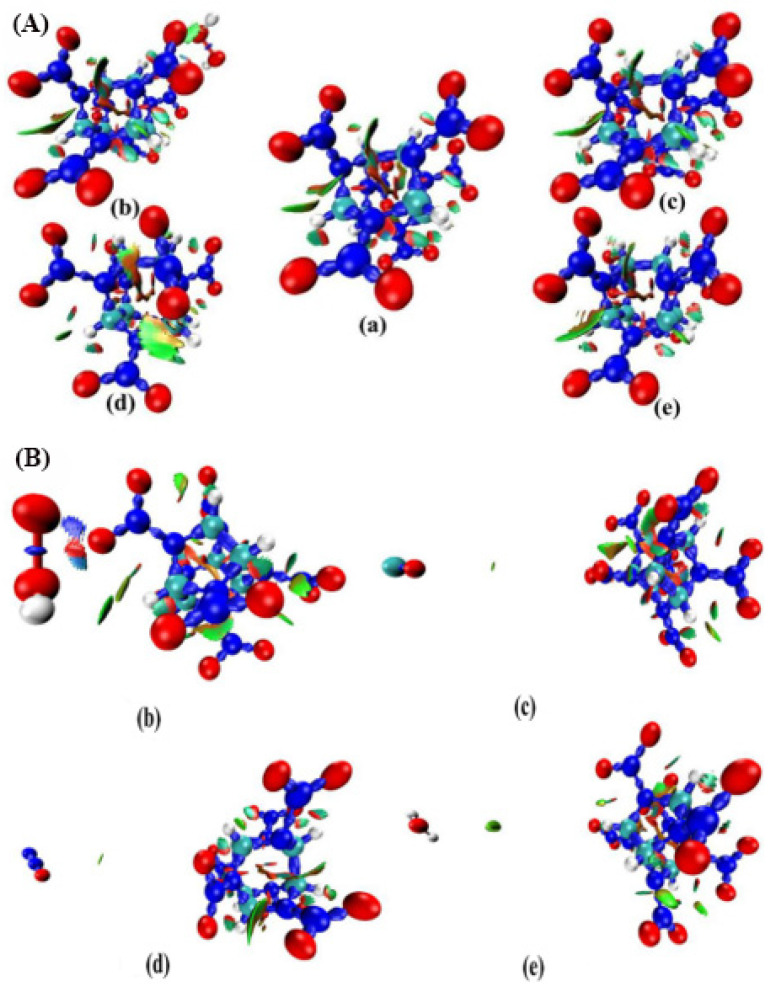
Plots of NCI isosurface for CL-20 and its host—guest complexes: (**a**) CL-20, (**b**) CL-20/H_2_O_2_, (**c**) CL-20/CO_2_, (**d**) CL-20/N_2_O, and (**e**) CL-20/H_2_O. The surfaces between the host and guest molecules correspond to the isosurface of RDG (isovalue = 0.5) mapped by sign (λ_2_)ρ function. The color scale is given in a.u.: (**A**) the interactions in the cage structure are highlighted, (**B**) the interactions between host and guest molecules are highlighted.

**Figure 11 molecules-27-03266-f011:**
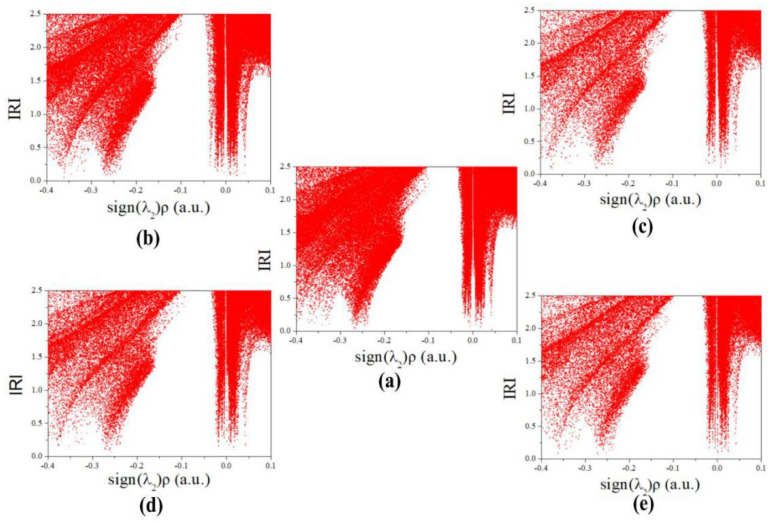
Plots of IRI for CL-20 and its host—guest complex: (**a**) CL-20, (**b**) CL-20/H_2_O_2_, (**c**) CL-20/CO_2_, (**d**) CL-20/N_2_O, and (**e**) CL-20/H_2_O.

**Figure 12 molecules-27-03266-f012:**
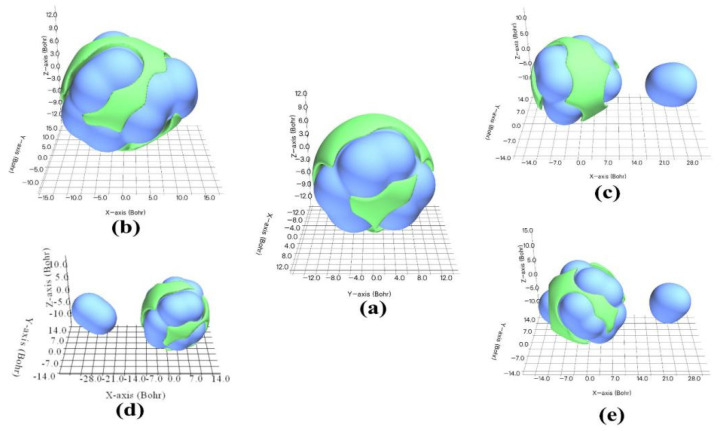
An amount of 0.5 kcal/mol isosurface of vdW potential of CL-20 and its complexes: (**a**) CL-20, (**b**) CL-20/H_2_O_2_, (**c**) CL-20/CO_2_, (**d**) CL-20/N_2_O, and (**e**) CL-20/H_2_O, respectively. Green and blue surfaces correspond to negative and positive parts, respectively.

**Figure 13 molecules-27-03266-f013:**
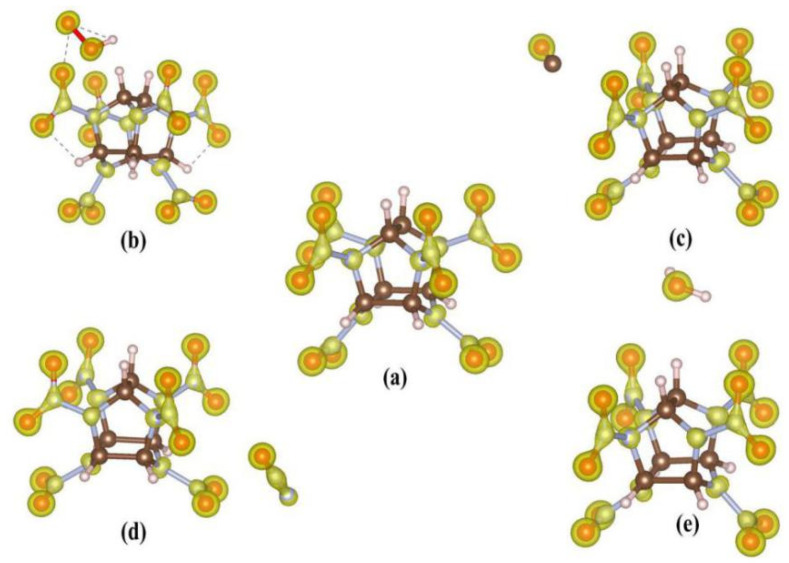
The electric charge density of CL-20 and its complexes: (**a**) CL-20, (**b**) CL-20/H_2_O_2_, (**c**) CL-20/CO_2_, (**d**) CL-20/N_2_O, and (**e**) CL-20/H_2_O, respectively.

**Figure 14 molecules-27-03266-f014:**
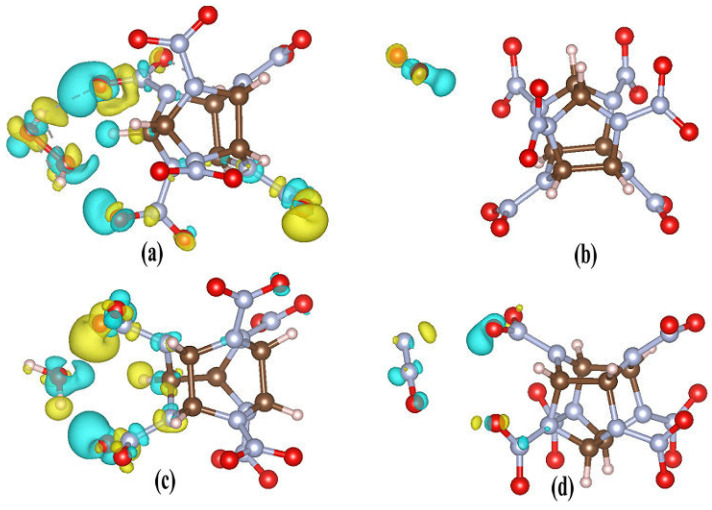
The differential charge density of CL-20 and its complexes: (**a**) CL-20/H_2_O_2_, (**b**) CL-20/CO_2_, (**c**) CL-20/H_2_O, and (**d**) CL-20/N_2_O, respectively. The isosurface value is set to 0.006 e·Å^−3^. Yellow represents charge accumulation.

**Figure 15 molecules-27-03266-f015:**
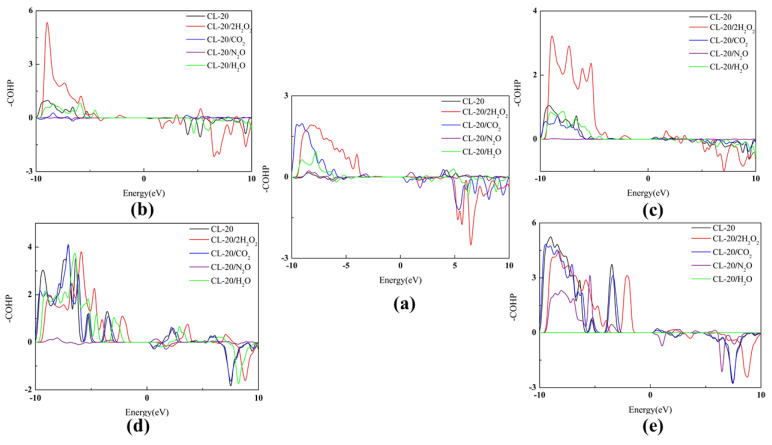
The -COHP curves for the CL-20 interactions of CL-20 and its complexes: (**a**) C-H bonds, (**b**) C-C bonds, (**c**) C-N bonds, (**d**) N-N bonds, and (**e**) N-O bonds, respectively. In the -COHP curves, the positive and negative signs represent bonding and antibonding states, respectively. The thick trendline corresponds to the actual concentration data of the corresponding matching color.

**Table 1 molecules-27-03266-t001:** The electrostatic interaction energy (kJ·mol^−1^) between the two fragments of four host–guest explosives.

Electrostatic Interaction Energy	CL-20/N_2_O	CL-20/H_2_O_2_	CL-20/H_2_O	CL-20/CO_2_
Guest…H	−0.21	−1.55	−0.45	−0.11
Guest…H	−0.23	−1.20	−0.47	−0.07
Guest…H	−0.23	−3.77	−0.29	−0.09
Guest…H	−0.28	--	--	−0.07
Guest…C	−0.13	−1.14	−0.27	−0.07
Guest…C	−0.14	−0.80	−0.24	−0.04
Guest…C	−0.13	−1.44	−0.26	−0.05
Guest…C	−0.18	--	--	−0.05
Guest…N	−1.4	−6.20	−1.47	−0.34
Guest…N	−0.96	−5.08	−1.42	−0.28
Guest…O	0.49	5.41	1.01	0.13
Guest…O	0.52	1.76	0.93	0.18
Guest…O	0.83	−0.65	0.9	0.33
Guest…O	0.7	1.81	0.84	0.38
Guest…N	0.34	2.62	0.42	0.08
Guest…N	0.31	1.36	0.43	0.13
Guest…CL-20	−0.14	−8.93	−0.11	0.03

**Table 2 molecules-27-03266-t002:** The charge of all atoms for CL-20 and its complexes.

	CL-20	CL-20/H_2_O_2_	CL-20/CO_2_	CL-20/N_2_O	CL-20/H_2_O
C1	3.439943	3.39528	3.365282	3.415394	3.484355
C2	3.368974	3.347578	3.463434	3.414919	3.389329
C3	3.407468	3.389311	3.438538	3.37998	3.440836
C4	3.361728	3.364322	3.42262	3.44343	3.464925
C5	3.426331	3.465616	3.465766	3.414552	3.449965
C6	3.466851	3.392609	3.49393	3.406848	3.448265
H1	0.760157	0.773847	0.779039	0.804121	0.769744
H2	0.764535	0.816269	0.724538	0.778757	0.782522
H3	0.764828	0.764591	0.727955	0.785294	0.767634
H4	0.788069	0.769692	0.76251	0.753908	0.70013
H5	0.735678	0.757301	0.748393	0.773186	0.784986
H6	0.774061	0.812159	0.735549	0.79133	0.763559
N1	4.30006	5.658855	5.674066	4.356636	5.641745
N2	5.73423	5.769231	4.295426	5.628674	5.661261
N3	4.292966	5.749727	5.641763	5.628986	5.688769
N4	5.691062	5.615677	4.311871	4.260681	5.666697
N5	4.307686	5.61475	5.724673	5.699055	5.638395
N6	5.671315	5.708823	4.291829	4.263904	5.622085
N7	4.335934	4.317027	5.75138	5.672504	4.344943
N8	5.758303	4.307445	4.228027	4.313575	4.308734
N9	4.273575	4.302678	5.659884	5.631423	4.276512
N10	5.641856	4.291919	4.290824	4.28659	4.314033
N11	4.276282	4.279109	5.638803	5.749158	4.29891
N12	5.640089	4.222286	4.324952	4.282773	4.288023
O1	6.420255	6.447034	6.427327	6.423941	6.370973
O2	6.404339	6.413201	6.425237	6.379549	6.403641
O3	6.442006	6.401898	6.422331	6.415535	6.399926
O4	6.425998	6.396192	6.400269	6.441002	6.428866
O5	6.448879	6.408842	6.423134	6.443255	6.428065
O6	6.39755	6.432817	6.437646	6.439187	6.442918
O7	6.397592	6.456496	6.437232	6.432066	6.447627
O8	6.383578	6.406194	6.444439	6.41519	6.39301
O9	6.407534	6.391632	6.419992	6.427163	6.410583
O10	6.452038	6.43961	6.408803	6.41524	6.411922
O11	6.418675	6.485976	6.393027	6.431183	6.442122
O12	6.41958	6.441443	6.404066	6.401798	6.438503

**Table 3 molecules-27-03266-t003:** The ICOHP between two atoms for CL-20 and its complexes.

		CL-20	CL20/2H_2_O_2_	CL-20/CO_2_	CL-20/N_2_O	CL-20/H_2_O
C2	C4	0.00002	−8.21658	−0.00173	−0.00057	0.0002
C1	C5	−0.00025	−8.03599	0.00198	−0.0002	0.01161
C3	C6	−6.7622	−8.14678	−0.04313	−0.01217	−6.60075
C4	N6	−0.00005	−9.51721	−0.00181	−0.00003	−0.0121
C5	N8	−0.00019	−9.71781	−0.00451	−0.00019	−0.01402
C3	N4	−0.00137	−9.55936	−0.00488	−0.00008	−0.01631
C5	N10	−0.0131	−9.69842	−0.01563	−0.00046	−0.01388
C4	N8	−0.01183	−9.79839	−0.01508	−0.00517	−0.01563
C2	N2	−0.01704	−9.78396	−0.01586	−0.00055	−0.01549
C6	N12	−0.01232	−9.96269	−0.01587	−0.00344	−0.0268
C3	N6	−0.00257	−10.05424	−0.0182	−0.04103	0.0168
C6	N10	−0.0202	−10.20592	−0.02711	−0.00554	−0.02778
C1	N12	−9.58089	−10.20256	−0.03043	−0.00211	−9.66028
C2	N4	−9.77474	−10.50934	−9.53927	−0.00988	−9.71233
C1	N2	−10.18768	−10.55467	−10.04402	−0.12441	−10.00431
H7	C4	0.0039	−6.82756	−0.00003	0.00352	0.00254
H9	C5	0.00039	−6.79854	−0.00024	−0.02047	0.00284
H11	C6	0.00094	−7.0202	−0.0017	−0.02405	−0.02669
H1	C1	−0.0436	−6.84098	−8.09852	−0.0435	−0.03322
H3	C2	0.00244	−6.83071	−7.98601	−0.46923	0.01673
H5	C3	−0.00919	−6.8202	−8.1083	−0.46346	−7.9553
N12	N11	−9.65004	−9.25005	−9.54528	−0.00024	−0.00058
N10	N9	−9.80733	−9.95989	−9.54889	−0.00002	0.00034
N2	N1	−10.25476	−9.99392	−9.97115	−0.00565	−0.00456
N8	N7	−10.43611	−10.66563	−10.11909	−0.01187	−0.00033
N6	N5	−10.92593	−11.20556	−10.81519	−0.4354	−0.00171
N4	N3	−11.03219	−11.40287	−11.25665	−0.4642	0.00082
N5	O5	−16.25535	−15.98383	−16.24132	0.00001	−9.41873
N7	O7	−16.5398	−16.29451	−16.47698	0.00006	−9.81729
N3	O3	−16.50557	−16.88862	−16.47518	−0.04298	−9.78804
N9	O9	−16.61515	−16.88697	−16.73727	−10.6239	−10.36198
N11	O11	−16.6889	−16.98825	−16.59197	−10.69822	−10.91247
N1	O1	−16.65992	−17.23416	−16.77597	−20.88737	−11.55274

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
