# Peer review of "Theoretical Studies on the Role of Guest in α-CL-20/Guest Crystals"

_molecules, 2022, doi:10.3390/molecules27103266_

Round 1

Reviewer 1 Report

REVIEW

Paper Id: molecules-1632610

Theoretical studies on the role of guest in α-CL-20/guest crystalsMingming Zhou1 and Caichao Ye2,Dong Xiang1*

Mingming Zhou and Caichao Ye, Dong Xiang

            This paper discusses the intermolecular interaction in the host-guest crystal structures of CL-20 in comparison to pure crystal structure of CL-20. I see the potential and motivitain behind these analysis, however quality of it is often difficult to judge. Some clarification and attention is needed for description of the structures and intramolecular interaction. I recommend that this paper be accepted after VERY major revision.

Primary concerns expressed were that:

  • I am missing crucial information which structures are used for the purpose of this analysis. If the influence of the incorporation of small guest is studied the crystal structures should be isostructural. Is it the case for pure CL-20 and host-guest system? Are they similar, as the crystal structures are not described at all it is very difficult to follow further discussion.
  • Author used Gaussian16 for the geometry optimization of what? Single molecule, dimer, crystal structures? Why do you need geometry optimization?
  • Are these structure of a good quality, are any disorder is present. Are the occupancy of the guest molecule equal to one? All further analysis depends on the quality of crystal structures. We should avoid “garbage in garbage out” There is no clear definition which structures where used, not even a picture showing them.
  • Section 3.1. Is the electrostatic potential map calculated for single molecule, dimer or in the crystal structures. Generally I often cannot find this information what part of crystal structures was used for the theoretical investigations. Why the distance between CL-20 are so different among the structures. The closest neighbors should be presented at the picture.
  • On Figure 2 we can see the closest contacts between host and guest. Are the CO2 really that far apart from CL-20? Why nitrogen atom from N2O is not incorporated in the Hirshfeld surface?
  • Figure 3 b, c, d have a long spikes that extend to very low values of de and di. Is the fingerprint plots calculated for single CL-20 ( I think that would be a better choice for a comparison) or for host-guest dimers? However the comparison with pure CL-20 would be difficult than. Are these calculation based on the optimized geometry or as it is from crystal structure determination?
  • Why author show only electrostatic interaction energy? How was it calculated?
  • As I am missing crucial information on crystal structure used and what part of the crystal structure is used for each calculations I cannot judge the analysis of the non covalent inetarctions.
  • Why electrostatic potential is analyzed twice?

Minor concerns were:

- lines 34-35 “The host-guest inclusion strategy is also solved the problem of time-consuming and

difficult by development of new energetic materials possessing high energy levels and

low sensitivities[3-7]. Need to be rewritten

- line 44 what is it HNIW?

Reviewer 2 Report

This manuscript demonstrates intermolecular interactions between Hexanitrohexaazaisowurtzitane (CL-20) and some small guest molecules evaluated by computational methods including DFT calculations and Hirshfeld surface analyses. However, insufficient information on detailed conditions for calculations is given. I think the lack of information hardly makes me assess the conclusions drawn by the authors. At least the geometrical parameters for Hirshfeld surface analyses should be given as supporting information. Were these host-guest co-crystal structures a priori derived by first-principles calculations, or were these obtained by exchanging guest molecules and local structural optimization based on the structure of known solvated crystals?

Author Response

This manuscript demonstrates intermolecular interactions between Hexanitrohexaazaisowurtzitane (CL-20) and some small guest molecules evaluated by computational methods including DFT calculations and Hirshfeld surface analyses. However, insufficient information on detailed conditions for calculations is given. I think the lack of information hardly makes me assess the conclusions drawn by the authors. At least the geometrical parameters for Hirshfeld surface analyses should be given as supporting information. Were these host-guest co-crystal structures a priori derived by first-principles calculations, or were these obtained by exchanging guest molecules and local structural optimization based on the structure of known solvated crystals?

Answer: The sufficient information on detailed structure information for calculations is given. The information structures for geometry optimization add the sentence “The process of geometry optimization is the process of obtaining reasonable structure. In order to obtain the reliable WFN file which was convert by CIF file format.” in line 91-93. The information structures for intermolecular contacts and electrostatic and vdW interaction characteristics of host and guest molecules add the sentence “During the theoretical investigations by CrystalExplorer, the chosen part of crystal structures is picked out automatically by CrystalExplorer software automatically from the CIF files. The chosen part of crystal structures is periodic for the crystal structures.” in line 110-113. The information structures for the effect on the chemical-bonding of CL-20 by the small molecules add the sentence “During the theoretical investigations by VASP, the chosen part of crystal structures is picked out by the nearest two neighbors.” in line 117-118.

The geometrical parameters for the host-guest co-crystal will be added in line 71-74 of the text shown as Figure 1. The calculation for Hirshfeld surface is based on the chosen part of crystal structures by CrystalExplorer software from the CCDC information. So, when the information of the host-guest co-crystal structures is added in the text, the geometrical parameters for Hirshfeld surface analyses is part of crystal structures.

The host-guest co-crystal structures information is from the CCDC.

Reviewer 3 Report

Zhou et al. provide in-depth theoretical insight and comparison of the intermolecular interactions and chemical bonding analysis of the high-energy explosive CL-20 and CL-20 based host-guest explosives. The paper is an important addition to the literature providing insights into the relatively few studies devoted to exploring more host−guest energetic complexes by introducing small molecules.  Although the authors do not make this point, this direction of research may eventually provide a theoretical basis for the synthesis of new CL-20-based inclusion compounds. As a result, it is worth publication and Molecules is an appropriate journal for this paper. However, some issues should be dealt with before publication. These are important, but relatively minor because they deal mostly with the presentation of the results, not the calculations themselves.

Firstly, In Figure 5 authors depict an ESP-mapped molecular vdW surface of (a) CL-20, (b) CL-20/H2O2, (c) CL-20/CO2, (d) CL-219 20/N2O, (e) CL-20/H2O. However, the value of local minima and maxima of ESP should be labeled in a bigger font, ideally close to the font size of the Figure caption. The global minimum and maximum which are labeled as blue and red are also hard to read, I suggest they make them bigger.

Secondly, Figure 7 to needs improvement. It fails to convey the message it wanted. First, the value and X and Y-axis in not readable, use a bigger font size. The labeling of the Figure is also inconsistent. While the X-axis in Figures 7a and 7e is bold, Figures 7b,d, and Figure 7c are not. This figure needs to be revised as suggested.

Third, In section 3.3. Effect on the chemical bonding of CL-20 by the small molecules, in lines 354-356 (The O atoms of the nitro branch chain have most electrons…..more electrons) looks confusing. Is this line correct?

Finally, the authors talked about the detonation performance of CL-20/H2O2 (on Page 4, line 151) but didn’t provide any results about the detonation velocity and detonation pressure of CL-20 based host−guest energetic materials discussed in this paper. 

Round 2

Reviewer 1 Report

I am satisfied with author responses.

Author Response

Getting your affirmation is the happiest thing for me. Hard work! Thank you very much!

Reviewer 2 Report

This manuscript demonstrates intermolecular interactions between Hexanitrohexaazaisowurtzitane (CL-20) and some small guest molecules evaluated by computational methods including DFT calculations and Hirshfeld surface analyses. This article will be worth being published in Molecules after minor revisions pointed as follows:

1. Showing a chemical digram of CL-20 will help readers a better understanding.

2. Red and blue spheres in Figure 2 are hardly recognized in their color. Drawing larger size ones will be required.

3. In Figure 4, sharp edge areas of the fingerprint plots in the small d(e)-d(i) region are important because these indicate short intermolecular contacts. Do these areas (two sets of edge in plots (a) to (d)) correspond to H-O contacts? In plot (e), additional two sets of edges appear in very short d(e)-d(i) areas (about 0.7 Å) which are not assigned. Please indicate the assignment of the element pair for the intermolecular contacts.
